# Neuroendocrine Neoplasms of the Breast: The Latest WHO Classification and Review of the Literature

**DOI:** 10.3390/cancers14010196

**Published:** 2021-12-31

**Authors:** Yukinori Ozaki, Sakiko Miura, Ryosuke Oki, Teppei Morikawa, Keita Uchino

**Affiliations:** 1Department of Medical Oncology, NTT Medical Center Tokyo, Tokyo 141-8625, Japan; tora02514@gmail.com (R.O.); keitauch0711@gmail.com (K.U.); 2Department of Breast Medical Oncology, The Cancer Institute Hospital of Japanese Foundation for Cancer Research, Tokyo 135-8550, Japan; 3Department of Diagnostic Pathology, NTT Medical Center Tokyo, Tokyo 141-8625, Japan; s.tazawa0330@gmail.com (S.M.); teppeimorikawa@gmail.com (T.M.); 4Department of Medical Oncology, The Cancer Institute Hospital of Japanese Foundation for Cancer Research, Tokyo 135-8550, Japan

**Keywords:** neuroendocrine carcinoma of the breast, neuroendocrine tumor, neuroendocrine carcinoma, review, classification, management

## Abstract

**Simple Summary:**

Breast tumors exhibiting neuroendocrine differentiation are a heterogeneous group of tumors that have been variously defined in previous World Health Organization (WHO) classifications. In the WHO Classification of Tumours, 5th edition, neuroendocrine tumors (NETs) and neuroendocrine carcinomas (NECs) of the breast, both of which are invasive cancers, are classified as neuroendocrine neoplasms (NENs) of the breast. However, the clinical significance of NE differentiation in breast cancers, especially in NETs of the breast, is not yet fully understood, and a large overlap appears to exist between breast cancers showing NE differentiation and invasive breast cancer of no special type (IBC-NST). While breast NECs show distinct clinical and morphological features, diagnosis of NETs based on the morphological characteristics alone can be challenging; one reason is that breast NETs do not necessarily have the same morphological characteristics as those of NENs arising in other organs. Thus, the heterogeneity of breast tumors with neuroendocrine differentiation and the changes in their classifications over the years have left many open issues that still need to be resolved. In this review, we shall summarize the history of breast “NENs,” including of mixed types of tumors and the characteristics of these tumors, and discuss their differences from NENs arising in other organs.

**Abstract:**

Breast tumors with neuroendocrine (NE) differentiation comprise an uncommon and heterogeneous group of tumors, including invasive breast cancer of no special type (IBC-NST) with NE features, neuroendocrine tumors (NETs), and neuroendocrine carcinoma (NEC). The most recent World Health Organization (WHO) classification in 2019 defined neuroendocrine neoplasms (NENs) of the breast (Br-NENs) as tumors in which >90% of cells show histological evidence of NE differentiation, including NETs (low-grade tumors) and NEC (high-grade). Due to the low prevalence of these tumors and successive changes in their diagnostic criteria over the years, only limited evidence of these tumors exists, derived mainly from case reports and retrospective case series. Breast tumors with NE differentiation are usually treated like the more commonly occurring IBC-NSTs. Immunohistochemistry (IHC) of breast tumors with NE differentiation usually shows a hormone receptor (HR)-positive and human epidermal growth factor type 2 (HER2)-negative profile, so that hormonal therapy with cyclin-dependent kinase (CDK)4/6 inhibitors or other targeted agents would be reasonable treatment options. Herein, we present a review of the literature on breast tumors with NE differentiation as defined in the latest WHO 2019 classification, and discuss the clinical management of these tumors.

## 1. Introduction

Neuroendocrine neoplasms (NENs) comprise a rare heterogeneous group of tumors defined as epithelial neoplasms composed of cells showing predominant neuroendocrinal (NE) differentiation and the characteristics of hormone-producing endocrine cells and nerve cells. NENs which possess neurosecretory granules in the tumor cells produce higher amounts of hormones than normal cells, which may cause clinical symptoms. These tumors can arise in any organ, including the intestine, pancreas, lung, and breast. Their clinical behaviors vary widely depending on the degree of NE differentiation of the tumor.

NENs have been classified in different and complex ways, depending on the location, morphology, proliferative activity, and hormonal activity. Functional NENs produce excessive amounts of hormones that cause clinical symptoms such as diarrhea and facial flushing. Non-functional NENs do not produce hormones in sufficient amounts to cause these symptoms; the majority of NENs are non-functional.

The major sites of NENs are the gastrointestinal tract, lung, and pancreas. According to one report, tumors in the gastroenteropancreatic system account for the majority (70%) of reported NENs, while those in the lung account for 25% [1]. A common classification framework for NENs was proposed by the World Health Organization (WHO) in 2018 [2], and the features of morphological differentiation, grade, proliferative activity, and extent of local spread, defined in the WHO classification of tumors of the gastroenteropancreatic system, are common to classification of NENs arising in any organ system [3].

NENs of the breast (Br-NENs) are the rarest of all NENs (accounting for less than 1% of all NENs) [4,5,6], and comprise a heterogeneous group of tumors. All Br-NENs are invasive carcinomas, by definition. The precise incidence rate remains uncertain, since immunohistochemical staining for Br-NENs is not conducted routinely, and the WHO classification of these tumors has changed several times even within the last decade. Therefore, there is no consensus on the clinical significance, treatment strategy, or prognosis of Br-NENs. Herein, we review the literature on Br-NENs, including mixed-type tumors, and the historical transition of the WHO classifications.

## 2. Methods

We conducted a comprehensive literature review of the PubMed database using terms of “breast” AND (“neuroendocrine carcinoma” OR “neuroendocrine tumors” OR neuroendocrine differentiation”) within the last 10 years, where the abstracts were available, in December 2021. Due to the limited evidence on treatment strategies, we conducted a hand search and reviewed additional literature (Figure 1).

## 3. WHO 2019 Classification

### 3.1. WHO Classification: Historical Transition

NE differentiation in breast carcinomas was first described in a case of mucinous carcinoma in 1963 [7]. In 1977, the first case series of eight cases of primary carcinoid of the breast was reported. These tumors were rich in argyrophilic granules and morphologically similar to carcinoid tumors in other organs [8]. In 1989, Papotti analyzed specimens from 100 cases of breast carcinoma using Grimelius silver staining, IHC for chromogranins A (CGA) and B and synaptophysin (SYN), and autoradiography with radiolabeled somatostatin. Of the 100 tumors, nine (9%) showed positive IHC for SYN, and four (4%) also showed positive IHC for chromogranins [9]. Bogina et al. evaluated the expressions of SYN and CGA by IHC in whole tissue sections of 1232 consecutive cases of invasive breast cancer. A total of 128 cases (10.4%) showed positive IHC for NE antigens; the staining was diffuse (>50% of the tumor cells) in 84 (6.8%) cases, and focal (10–49% of tumor cells) in 44 (3.6%) cases. Of these 128 breast cancers, 95 were IBC-NSTs, 5 were invasive lobular carcinomas, 7 were mucinous carcinomas, and 21 were solid papillary carcinomas (SPCs) [10].

In the WHO classification 3rd edition (2003), NEC of the breast was classified as a separate entity for the first time, and was defined according to the criteria proposed by Sapino et al. (Figure 2) [11]. Neuroendocrine tumors (NETs) were defined as epithelial tumors having similar morphologies to NETs of the gastrointestinal tract and lung, with more than 50% of the tumor cells expressing NE markers (CGA and SYN). In the 4th edition (2012), the basic concept of classifying primary breast NENs as NET and NEC was established, as in other organs. Br-NENs were defined as those showing similar morphological characteristics to those of gastrointestinal tract and pulmonary NETs, expressing NE markers to a greater or lesser degree, and classified into two major categories: “NET, well-differentiated” and “NEC, poorly differentiated/small cell carcinoma.” The 4th edition also recognized a third category: a subset of breast cancers showing NE differentiation.

In regard to Br-NENs, the current WHO classification of Tumours 5th edition (2019) describes the need to refine and improve the diagnostic reproducibility of these tumors, clarify their clinical significance, and follow the recent unified classification of NENs arising other organs. As in the previous editions of the classification, the 5th edition also classifies NETs as well-differentiated NENs, and NECs as a poorly differentiated NENs, exhibiting the histological and immunohistochemical characteristics of NE differentiation. The main changes in the latest WHO classification regarding NENs are the listing of the tumor types according to the percentage of cells showing NE differentiation in the tumors, exclusion of particular histologic types (solid papillary carcinoma and hypercellular variants of mucinous carcinoma), and inclusion of large cell neuroendocrine carcinoma (LCNEC). Parameters used in other organ systems, such as the mitotic count, Ki-67 index, presence/absence of necrosis, and expression profile of peptide hormones, are not applicable to classification of NENs of the breast. Since most Br-NENs, regardless of the malignancy grade, show an admixture of elements of both neuroendocrine tumors and invasive cancer, the following recommendations have been made: invasive carcinoma with more than 90% of cells exhibiting NE morphology be classified as NET or NEC, those with 10–90% of cells exhibiting NE morphology be classified as mixed invasive NST and NET/NEC, and those with less than 10% of cells exhibiting NE differentiation be classified as invasive carcinoma of NST, with an option for comment on the focal NE pattern [12] (Figure 2).

### 3.2. Histology: Neuroendocrine Tumors, Neuroendocrine Carcinoma

In the WHO Classification of Tumours 5th edition, NECs are characterized as a high-grade malignant tumor with histological features similar to those of small cell neuroendocrine carcinoma (SCNEC) and LCNEC of the lung (Figure 3). Although it is a rarely encountered tumor, it exhibits distinct morphologic features.

The morphological characteristics of NETs remain unclear. NETs are defined as invasive tumors of low/intermediate-Nottingham grade with NE differentiation, supported by the presence of neurosecretory granules and diffuse, uniform immunoreactivity for NE markers (Figure 4). In addition to traditional NE markers, INSM1 is reported as a novel marker and could have a role in diagnosing of NENs [13]. Histologically, NETs are characterized by densely cellular, solid nests and trabeculae of cells varying from spindle-shaped/plasmacytoid/polygonal cells with eosinophilic and granular cytoplasm to large clear cells within a delicate fibrovascular stroma. NETs of the breast do not necessarily exhibit such features as ribbons, cords, and rosettes, which are classic features of carcinoid tumors of the lung or NETs in the gastroenteropancreatic system [12]. Solid papillary carcinoma and hypercellular mucinous carcinoma with expression of NE markers are now considered as distinct diseases and excluded from the current classification of NETs. In the 5th edition of WHO classification, it is also stated that NETs should be distinguished from other histological types of breast cancer expressing NE markers by the presence and extent of histological features characteristic of NE differentiation of the tumor [12].

According to one previous report, breast tumors with NE differentiation are underrecognized and difficult to diagnose. They often lack the characteristic nuclear findings seen in NENs arising in other organs. In addition, some primary breast tumors with NE differentiation may resemble invasive ductal carcinoma, invasive lobular carcinoma, or even DCIS. Metastases from other organs can also be diagnosed as primary breast NENs. The authors conclude that careful observation of the cytologic and structural features and confirmation by IHC are helpful for accurate classification of breast tumors with NE differentiation [14].

The morphological features of breast carcinoma with NE differentiation have also been studied [14,15,16]. In one study, the characteristics of tumors in which more than 50% of the cells were found to show positive IHC for NE markers were compared with those of tumors that were morphologically similar, but showed negative IHC for NE markers. The authors noted the presence of large-sized, solid, cohesive nests, intermediate nuclear and histologic grades, plasmacytoid, spindle shape, and/or columnar shape of the tumor cells, an eosinophilic-granular appearance of the cytoplasm, and round nuclei as findings suggestive of a breast carcinoma with a NE component [16].

Most breast cancers are diagnosed by needle biopsy; however, other than the documented difficulty in diagnosing NE-differentiated breast tumors by needle biopsy [14], there is very little literature that discusses in detail the diagnostic characteristics of breast tumors with NE differentiation in needle biopsy specimens. Several cytologic features of NETs have been mentioned; NETs are characterized by loosely cohesive sheets of well-ordered cells with plasmacytoid, eccentric granular cytoplasm, round nuclei with speckled “salt and pepper” chromatin, and inconspicuous nucleoli. NEC resembles small cell NEC of the lung and NECs arising at other sites [17].

Some argue that treating Br-NENs, in particular, NETs, in the same way as NENs arising in other organs requires careful evaluation: current NETs of the breast do not exhibit a definite and unequivocally recognizable morphology and the diagnosis relies on IHC for CGA and SYN, which are also sometimes expressed in other non-NENs. In addition, the authors have pointed out that the clinical behaviors and responses to therapy of breast NETs overlap with those of non-NE breast carcinomas, and that molecular and genetic analyses have revealed more similarities of breast NETs to luminal A type breast cancer than to NETs arising in other organs [18]. Thus, further studies of breast tumors with NE differentiation are needed, including to better define and classify these tumors.

## 4. Clinical Features

The clinical presentation of patients with Br-NENs is not different from that of patients with IBC-NSTs. This disease is more commonly diagnosed in postmenopausal or older women [5]. There are no clinical reports of Br-NEN as manifesting with clinical syndromes related to ectopic production of any hormones, such as carcinoid syndrome. SCNECs have been reported to be diagnosed at a more advanced stage than other types of cancers [19], and the reported distant metastasis rate is 19–30% [20,21].

There are no specific radiological features of primary Br-NENs [22], and the diagnosis is confirmed on the basis of the histopathological findings. Gallo et al. recently reviewed the mammographic findings in case reports and case series of Br-NENs, and reported that the most common mammographic appearance was a hyperdense, irregularly shaped solitary mass without calcifications [23]. Computed tomography (CT) is performed to detect distant metastases and the possibility of the breast tumor being a metastasis from primitive NETs arising in other organs. It is reasonable to perform somatostatin receptor scintigraphy or positron emission tomography (PET)-CT with gallium-68-labeled somatostatin analogues to evaluate the disease location in cases of well-differentiated NETs and indication of peptide receptor radionuclide therapy (PRRT). PET-CT with 18-fluorodeoxyglucose can be performed in patients with poorly differentiated tumors or high-grade carcinomas [24].

## 5. Management

Due to the low incidence as well as their complexity, there are few reports of specific clinical trials for Br-NENs. There is limited evidence to recommend any treatment for patients with this disease, and Br-NENs are currently treated like any IBC-NST (Figure 5). The treatment strategy should be determined taking into consideration the prognostic or predictive factors, including the TNM stage, estrogen receptor (ER), progesterone receptor (PgR), HER2 status, histological grade, nuclear grade, Ki67-index, age, menopausal status, and general condition, as for other invasive breast cancers. Adjuvant therapy and radiation should be considered based on these clinicopathological features after surgery. It is reported that the efficacy of neoadjuvant chemotherapy and adjuvant chemotherapy are equivalent in the aspect of distant recurrence and breast cancer mortality in IBC-NSTs [25], and neoadjuvant therapy is administered as a standard treatment. In addition, in recent years, escalation or de-escalation treatment, in which postoperative treatment can be individualized according to the efficacy and response to neoadjuvant therapy, has become the standard strategy [26]. The efficacy of neoadjuvant therapy for Br-NENs is unclear, however, case series or case reports have shown that the efficacy of neoadjuvant chemotherapy for NECs of the breast [27,28,29]. Considering that NECs are often diagnosed at a relatively advanced stage, neoadjuvant therapy is a reasonable treatment strategy.

### 5.1. Surgery

Surgery is the recommended treatment for patients with resectable Br-NENs. It is important to differentiate between primary NENs and metastatic NE tumors from other organs to determine the optimal surgical approach. To diagnose the primary NENs, morphological findings, IHC markers including NE markers and hormone-receptors, intraductal lesion, clinical history, and clinical symptoms such as diarrhea and facial flushing should be considered [30]. There is little reported evidence for the optimal extent of resection for primary early Br-NENs [31]. Although breast-conserving surgery is a frequently adopted option for IBC-NSTs, mastectomy may be the preferred surgical option for NEC of the breast, due to the potentially aggressive nature of these tumors [32].

### 5.2. Radiotherapy

Radiotherapy for the chest wall and regional lymph nodes should be administered similarly to that for cases of IBC-NSTs [27,33,34]. There are no reports of specific clinical trials of radiotherapy after surgery for patients with Br-NENs. One case-controlled study showed a trend toward improved survival with radiotherapy in patients with NEC of the breast as defined in the WHO 2003 classification [35], whereas another population-based analysis showed that adjuvant radiotherapy was not associated with improved survival in patients with primary SCNEC of the breast [20].

### 5.3. Chemotherapy

Use of adjuvant systemic therapy should be decided based on the clinicopathological features and risk of recurrence in individual patients. Although there are no reports of specific clinical trials carried out to evaluate the efficacy of adjuvant chemotherapy in patients with Br-NENs, patients with high-risk disease should be offered adjuvant or neoadjuvant chemotherapy on an individual basis. The main factors considered for initiating patients with IBC-NSTs on adjuvant or neoadjuvant chemotherapy are the tumor size, nodal status, nuclear grade, age, tumor subtype based on the results of IHC for ER, PgR, and HER2, and the Ki67 index. The tumor size and nodal status are also the major predictors of recurrence in patients with Br-NENs [5,19]. Based on data of gastrointestinal NETs [36,37], NETs of the breast may be less sensitive to chemotherapy than IBC-NSTs. On the other hand, NECs arising in other organs, such as pulmonary or gastrointestinal NECs, are usually sensitive to chemotherapy. The use of adjuvant chemotherapy in patients with NECs is reasonable, but the optimal regimen has not yet been established. Use of chemotherapy regimens including platinum agents and etoposide or taxanes, as well as of an anthracycline plus taxane regimen for IBC-NSTs is commonly reported in the literature and case series [28,38,39]. The anthracycline plus taxane regimen is a reasonable treatment strategy for patients with high-risk Br-NENs and further clinical research is warranted.

The development of escalation therapy or response-guided treatment is being led mainly in triple-negative breast cancer (TNBC). Disease-free survival of the patients who had residual disease of triple-negative breast cancer after standard neoadjuvant chemotherapy has been improved by adjuvant treatment with capecitabine for six months [40]. Although there is no data to show the efficacy of capecitabine in Br-NENs, a similar strategy for HR-negative and HER2-negative Br-NENs could be considered based on the risk-benefit balance.

### 5.4. Hormonal Therapy

Br-NENs are commonly HR-positive, so that hormonal therapy is a reasonable treatment option, based on case reports [41,42]. Adjuvant hormonal therapy for HR IBC-NST is recommended for 5–10 years [43]. In premenopausal patients, TAM alone was common, but the combination with luteinizing hormone-releasing hormone analog is also a standard treatment depending on the risk of recurrence. Adjuvant hormonal therapy for 5–10 years is also considered reasonable for HR-positive Br-NEN. For metastatic BC, in pivotal clinical trials conducted in patients with HR-positive IBC-NSTs, combined therapy with a hormonal agent such as an aromatase inhibitor or fulvestrant, and a CDK4/6 inhibitor was shown to yield a significant improvement of the progression-free survival and overall survival compared with hormonal therapy alone [44,45,46,47,48]. Combined use of a CDK4/6 inhibitor with hormonal therapy could also be a useful strategy for Br-NENs, and a case report documents a durable response to palbociclib plus fulvestrant treatment in a patient with high-grade NEC of the breast who was refractory to platinum-based chemotherapy and other hormonal therapy [49]. Recently, abemaciclib has been approved as an adjuvant therapy for patients who had high-risk HR-positive HER2-negative IBC-NST [50], which could be considered also for high-risk HR-positive HER2-negative Br-NENs as a latest treatment strategy.

Recent molecular researches have revealed *PIK3CA* mutations in 7–33% of Br-NENs, although the frequency is lower than that reported in HR-positive HER2-negative IBC-NSTs [51,52,53,54]. Targeting the *PI3K*/*AKT*/*mTOR* pathway using a PI3K inhibitor (e.g., alpelisib) and mTOR inhibitor (e.g., everolimus) is a standard therapeutic strategy for HR-positive, HER2-negative IBC-NSTs. The SOLAR-1 trial reported alpelisib plus fulvestrant as being effective in patients with *PIK3CA*-mutant HR-positive HER2-negative IBC-NSTs who have previously received hormonal therapy [55]. Everolimus plus exemestane is established as the standard treatment for patients with HR-positive HER2-negative IBC-NSTs who have previously received treatment with an aromatase inhibitor [56], and everolimus has been demonstrated to be effective in patients with advanced pancreatic NETs [57]. Interestingly, oncogenic or likely oncogenic mutations of *PI3K* pathway was more common in NETs compared to NECs (50% vs. 18.2%) [58]. Therefore, targeting the *PI3K*/*AKT*/*mTOR* pathway may be a reasonable and promising strategy for treating HR-positive HER2-negative Br-NENs, especially for NETs (Figure 6).

### 5.5. Anti-HER2 Therapy

It is reasonable to use anti-HER2 therapy for HER2-positive Br-NENs. Although there is little evidence of the predictive role of the HER2 status in NENs, there is a case report in the literature of the efficacy of trastuzumab treatment in a patient of NEC with HER2 amplification of the breast [59]. Another case of well-differentiated NET of the breast, according to WHO 2012 classification, with HR-positive HER2-positive status received surgery, adjuvant chemotherapy, trastuzumab, and hormonal therapy, resulting in disease-free after nine years of follow-up [60]. Pertuzumab, another anti-HER2 agent, and trastuzumab-emtansine (T-DM1), an antibody-drug conjugate (ADC), have been established as a standard treatment for HER2-positive IBC-NSTs both in adjuvant setting and metastatic setting [61,62,63,64]. Tyrosine kinase inhibitors, including lapatinib, neratinib and tucatinib, have shown efficacy and been approved for HER2-positive IBC-NSTs [65,66,67]. Recently, a novel ADC, trastuzumab-deruxtecan, has been approved for HER2-positive IBC-NSTs [68]. Although there is limited data on the efficacy of these drugs against Br-NENs, the use of these drugs against HER2-positive Br-NENs can be considered.

### 5.6. Somatostatin Analogue

Somatostatin analogues are a very important therapeutic option in the diagnosis and treatment of NETs in the gastroenteropancreatic system. NETs of the breast have also been reported to be positive for somatostatin receptor types 2, 2A, 2B, 3 and 5 by IHC [69]. While it can be an effective treatment option, the treatment of somatostatin analogues has not shown clinical benefit for NETs of the breast at this time [70].

### 5.7. Peptide Receptor Radionuclide Therapy (PRRT)

PRRT with radiolabeled somatostatin analogues for somatostatin receptor-targeted PET-CT have shown efficacy for somatostatin receptor-expressing NENs and is expected to be an effective therapeutic strategy. There are two case reports, one with 90Y-DOTATOC and the other with 177Lu-DOTATOC, showing the promising efficacy of PRRT for NETs [71,72]. A phase II study is underway to evaluate the safety and efficacy of 177Lu-DOTATOC in somatostatin receptor-expressing NETs, including NETs of the breast (NCT04276597).

### 5.8. Other Agents

Immune checkpoint inhibitors are being developed for breast cancer and have already become the standard of care for PD-L1-positive TNBC. Since the efficacy of single agents has been inadequate, combination therapy with chemotherapy or other targeted drugs is now being developed. Atezolizumab and pembrolizumab have shown efficacy and manageable toxicity against PD-L1-positive metastatic TNBC in combination with chemotherapy [73,74]. There is no clinical data of immune checkpoint inhibitor for Br-NENs. PD-L1 status was evaluated in NENs originated from various organs, which showed that PD-L1 expression was detected in 31.6% of NECs and 0% of NETs [75]. Microsatellite instability and high mutational load are pronounced in NECs, which suggest that immune checkpoint inhibitors are promising therapeutic option in NECs [76]. Phase 1/2 studies were conducted to evaluate the role of immune checkpoint inhibitors in NENs, which showed promising efficacy and manageable toxicity [77]. It is reasonable to assess the PD-L1 status of HR-negative HER2-negative NEC of the breast to consider the use of immune checkpoint inhibitors.

Sacituzumab govitecan is an ADC consisting of a humanized immunoglobulin IgG1 monoclonal antibody (hRS7) targeting trophoblast cell surface antigen-2 conjugated to SN-38, an active metabolite of irinotecan, which has shown the efficacy in patients with TNBC who received standard chemotherapy [78]. Treatment with sacituzumab govitecan could be considered for HR-negative HER2-negative NEC that has progressed on standard chemotherapy.

## 6. Prognosis

There are conflicting data about the prognosis of breast cancers with NE differentiation, which could be because of the rarity of these heterogeneous tumors and changing classification criteria. In the WHO 2019 classification, solid papillary carcinoma and the hypercellular-subtype mucinous carcinoma were excluded from NENs, whereas these tumors were included in the WHO 2003 and 2012 classifications. These tumors carry a relatively better prognosis than other high-grade NENs, which could cause discrepancies in the reported prognoses [79].

A recent SEER database analysis including 361 cases of NENs included in the database between 2003 and 2016 showed a five-year disease-specific survival rate and five-year overall survival of NEC of the breast, as defined according to the WHO 2019 classification, of 46.0% and 38.87%, respectively [80]. In another report, SCNEC carried a worse prognosis than other NE tumors, with five-year disease-specific survival and overall survival rates of 50.5% and 32.2% in patients with the former, compared with to 74.0% and 62.4%, respectively, in patients with well-differentiated NE tumors [81]. Another review of 142 cases in the SEER database showed shorter overall survival and disease-specific survival rates in patients with NEC of the breast compared with those with IBC-NSTs. Furthermore, multivariate analysis identified NE differentiation as an independent poor prognostic factor (hazard ratio for disease-specific survival, 1.80; 95% CI: 1.36–2.37), in addition to patient age and tumor size, nodal status, histologic grade, and HR status [5].

In other organs than the breast, well-differentiated NETs usually carry a better prognosis than carcinomas without NE features, due to their slow-growing nature. However, there is no clear evidence to suggest that NETs of the breast carry a better prognosis than IBC-NST. A recent study of 1372 invasive breast cancers, which included 52 (3.8%) NETs, showed similar survival data between the NETs and invasive breast cancers without neuroendocrine features [82]. On the other hand, a recent SEER database analysis of 239 NETs and 491,908 of invasive ductal carcinomas of NST as defined according to the WHO 2019 classification revealed five-year disease-specific survival rates of 63.39% and 89.17%, respectively [80]. This analysis also showed that the prognosis of NETs and NEC of the breast were significantly worse than that of stage- as well as grade-adjusted IBC-NST. The Nottingham histological grade and tumor stage are reported as prognostic predictors in patients with NETs, as in those with IBC-NSTs [51,83]. Interestingly, low expression levels of CGA and SYN were associated with a poor prognosis in breast cancers with NE features, and the clinicopathological profile of tumors with CD56-only positivity differed from that of tumors showing CGA and SYN expressions. The impact of the NE expression level on the prognosis of NETs and invasive breast cancers with NE differentiation is still controversial, so that further research is warranted.

## 7. Conclusions

Breast tumors with NE differentiation comprise an uncommon and heterogeneous group of tumors that show histological overlap with invasive breast cancers of no special type. While NECs of the breast show distinct morphological and clinical features, diagnosis of NETs of the breast based on the morphological characteristics alone can be challenging. There is limited evidence to recommend any specific treatment strategies for breast cancers with NE differentiation, which are currently treated as invasive breast cancers of no special type. Further studies of breast tumors with NE differentiation are needed to better define and classify this group of tumors and to establish effective management strategies.

## Figures and Tables

**Figure 1 cancers-14-00196-f001:**
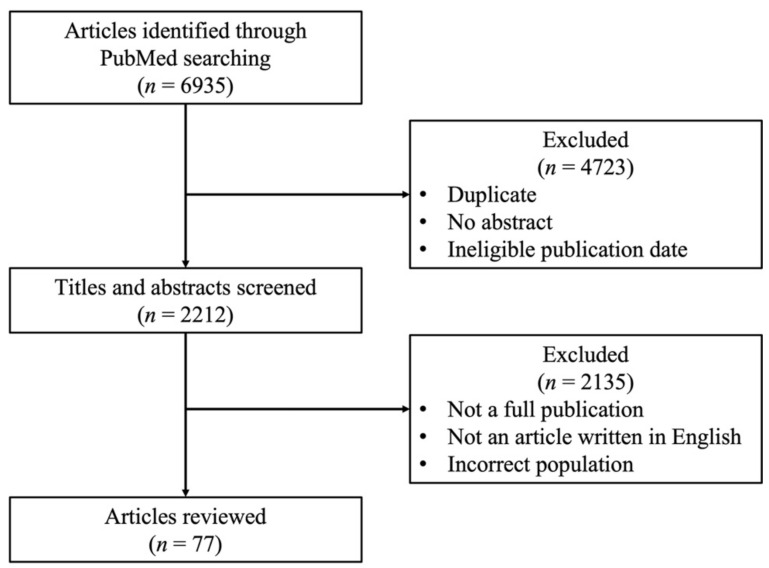
Literature flow chart of the PubMed database using terms of “breast” AND (“neuroendocrine carcinoma” OR “neuroendocrine tumors” OR “neuroendocrine differentiation”).

**Figure 2 cancers-14-00196-f002:**
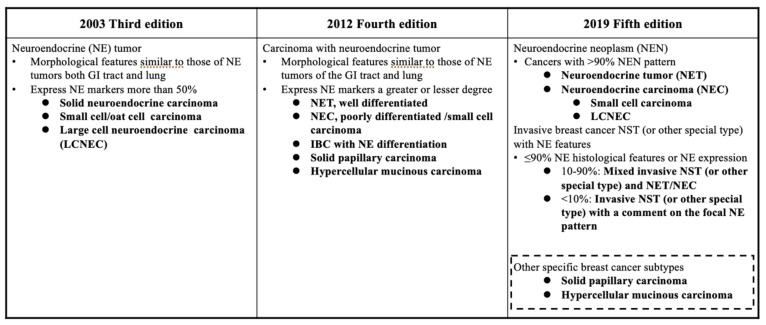
Summary of changes in the WHO Classification of NE tumors over the years (original figure by authors).

**Figure 3 cancers-14-00196-f003:**
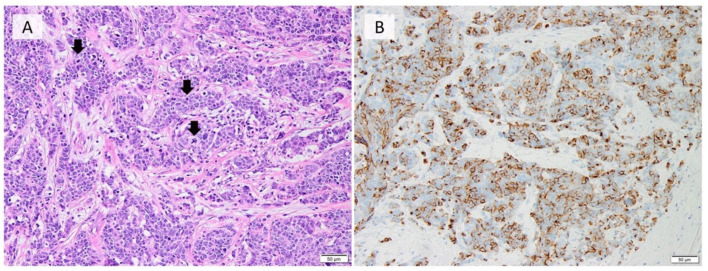
Morphological and immunohistochemical features of NEC of the breast. Neuroendocrine carcinoma (NEC) of the breast showing infiltrative, densely packed hyperchromatic cells with a high nuclear-to-cytoplasmic ratio. Mitotic figures are seen (arrows). (**A**) H & E staining (×200); (**B**) positive immunostaining for chromogranin A (×200) (Pictures by authors).

**Figure 4 cancers-14-00196-f004:**
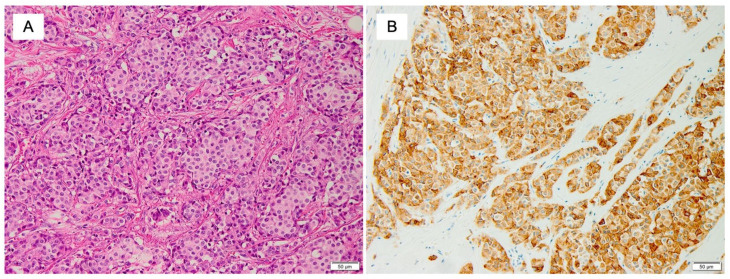
Morphological and immunohistochemical features of NET of the breast. Neuroendocrine tumor (NET) of the breast characterized by closely packed nests of cells with round and uniform nuclei. (**A**) H & E staining (×200); (**B**) positive immunostaining for synaptophysin (×200) (Pictures by authors).

**Figure 5 cancers-14-00196-f005:**
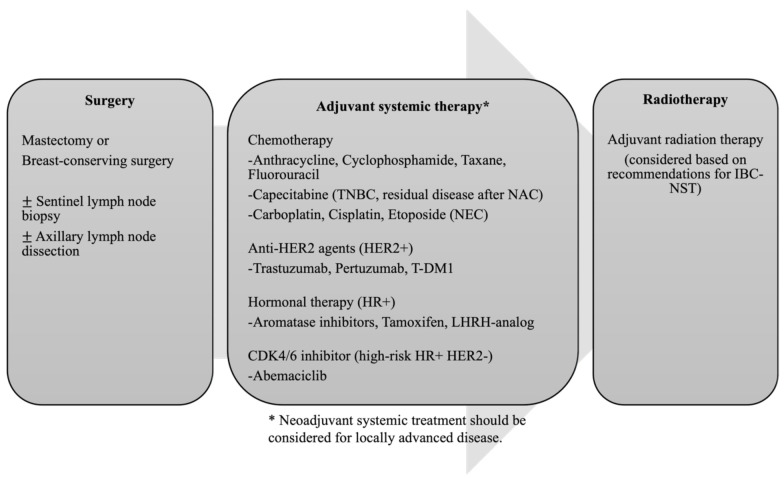
Overview of treatment strategy for resectable Br-NEN. Treatment strategy including surgery, adjuvant systemic therapy, and radiotherapy should be considered based on recommendations for IBC-NST. Neoadjuvant systemic therapy should be considered for locally advanced disease.

**Figure 6 cancers-14-00196-f006:**
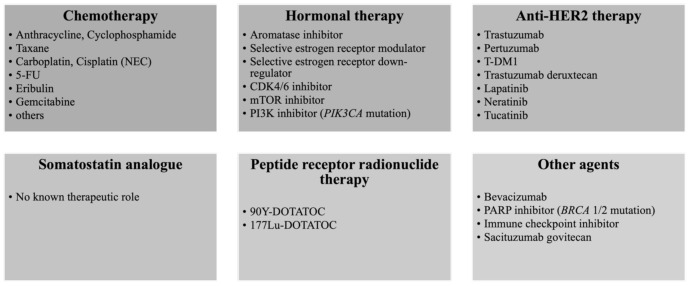
Systemic therapy for recurrent or metastatic Br-NEN. Systemic therapy for recurrent or metastatic Br-NEN including chemotherapy, hormonal therapy for HR-positive disease, anti-HER2 therapy for HER2-positive disease, somatostatin analogue, peptide receptor radionuclide therapy, and other agents.

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
