# Peer review of "Neuroendocrine Neoplasms of the Breast: The Latest WHO Classification and Review of the Literature"

_cancers, 2021, doi:10.3390/cancers14010196_

Round 1

Reviewer 1 Report

In this paper Authors performed a review of the literature on breast NENs. 

The topic is interesting due to the rarity of the disease but I have some concerns about how the authors manage the information obtained

First of all authors pointed out the topic on Br NEN disease but in the therapeutic section they underlined the role of treatments on breast cancer with neuroendocrine features that according the WHO classification is different from Br NETs for example

Consequentely no mention has been made about the role of PRRT that is a crucial point in the management of NENs

So authors should re-focus the therapeutic part and add information (if available) on Br NEN treatment above all PRRT and target therapy and so on.. ( i.e. doi: 10.1097/RLU.0000000000003005)

Author Response

Point 1: In this paper Authors performed a review of the literature on breast NENs. 

The topic is interesting due to the rarity of the disease but I have some concerns about how the authors manage the information obtained

Response 1: I sincerely appreciate the reviewer’s important comment. We have added method section and literature flow chart (Figure 1) to clarify how to obtain and review the literatures.

Point 2: First of all authors pointed out the topic on Br NEN disease but in the therapeutic section they underlined the role of treatments on breast cancer with neuroendocrine features that according the WHO classification is different from Br NETs for example. Consequentely no mention has been made about the role of PRRT that is a crucial point in the management of NENs

Response 2: Thank you for the comment. We have changed the management section to focus on Br-NEN disease. We have added the information on management, surgery, chemotherapy, hormonal therapy, anti-HER2 therapy, somatostatin analogue, peptide receptor radionuclide therapy (PRRT) and other agent section. Also Figure 5 and Figure 6 were added to show overview of the treatment strategy of Br-NEN.

Point 3: So authors should re-focus the therapeutic part and add information (if available) on Br NEN treatment above all PRRT and target therapy and so on.. ( i.e. doi: 10.1097/RLU.0000000000003005)

Response 3:  I appreciate your important suggestion. As mentioned in point 2, we have added the information on management section including PRRT, somatostatin analogue and other targeted agent section.

We hope that our revised manuscript will be considered for publication in the Cancers. We look forward to hearing from you in due course.

Yours sincerely.

Yukinori Ozaki

Reviewer 2 Report

In this paper by Yukinori Ozaki , the authors deal with Neuroendocrine neoplasms of the breast: the latest WHO classification and review of the literature.

Comments:

  • I suggest making a flowchart with a paper design
  • My main criticism is methodology. There is no information about this at all in the article. I am interested in how the authors selected the papers that were then analyzed. What databases they searched, what keywords were used.
    Maybe they rejected some articles? What criterion did they adopt regarding the year of publication that was taken into account? Such information should be included in the review article. Maybe a flow diagram will be a good option?
  • Make a scheme with the management

Author Response

Point 1: I suggest making a flowchart with a paper design. My main criticism is methodology. There is no information about this at all in the article. I am interested in how the authors selected the papers that were then analyzed. What databases they searched, what keywords were used. Maybe they rejected some articles? What criterion did they adopt regarding the year of publication that was taken into account? Such information should be included in the review article. Maybe a flow diagram will be a good option?

Response 1: I sincerely appreciate the reviewer’s important comment. We have added method section as shown below and literature flow chart (Figure 1) to clarify how to obtain and review the literatures.

We conducted a comprehensive literature review of the PubMed database using terms of “breast” AND (“neuroendocrine carcinoma” OR “neuroendocrine tumors” OR neuroendocrine differentiation”) within the last 10 years, which were available of abstract, on December 2021. Due to the limited evidence on treatment strategies, we conducted a hand search and reviewed additional literature (Figure 1).

Point 2: Make a scheme with the management

Response 2: Thank you for the important comment. We have added Figure 5 and Figure 6 to show overview of the management of Br-NENs as shown below.

We hope that our revised manuscript will be considered for publication in the Cancers. We look forward to hearing from you in due course.

Yours sincerely.

Yukinori Ozaki

Reviewer 3 Report

The review paper entitled “Neuroendocrine neoplasms of the breast: the latest WHO classification and review of the literature”, despite dealing with a topic of high clinical impact, it does not show elements of novelty compared to what is already present in the literature on the same topic.

Therefore, the data reported in this manuscript do not provide an added value compared to those published and still available. Indeed, right in 2021, respectively in January and October, two reviews were published on the same topic (https://pubmed.ncbi.nlm.nih.gov/33584543/; https://pubmed.ncbi.nlm.nih.gov/34698887/) and in particular the first article is very similar both in terms of structure and of contents, while the second, being published in October 2021, is even more updated. Strangely the authors did not cite either of the two articles.

Furthermore, another recent paper on the topic published in October 2021 was not cited by the authors (https://pubmed.ncbi.nlm.nih.gov/34728787/ ).

In addition, it is not clear if the images reported in the manuscript are from the research group of the authors or if they were obtained from the literature, as the authors have not even cited their articles published on this topic.

In my opinion the manuscript should be improved taking into consideration that a review paper must be updated with the latest articles published on the topic and the authors should explain what is the novelty of their manuscript in light of the similar articles mentioned above. In addition, the authors should specify if they have contributed data published by their group to the bibliographic review reported in the paper.

Author Response

Point 1: The review paper entitled “Neuroendocrine neoplasms of the breast: the latest WHO classification and review of the literature”, despite dealing with a topic of high clinical impact, it does not show elements of novelty compared to what is already present in the literature on the same topic. Therefore, the data reported in this manuscript do not provide an added value compared to those published and still available. Indeed, right in 2021, respectively in January and October, two reviews were published on the same topic (https://pubmed.ncbi.nlm.nih.gov/33584543/; https://pubmed.ncbi.nlm.nih.gov/34698887/) and in particular the first article is very similar both in terms of structure and of contents, while the second, being published in October 2021, is even more updated. Strangely the authors did not cite either of the two articles.

Response 1: I sincerely appreciate the reviewer’s important comment.  We totally agree with the reviewer’s comment. First, we have added method section as shown below and literature flow chart (Figure 1) to clarify how to obtain and review the literatures. Revised manuscript cites the two important reviews mentioned.

We conducted a comprehensive literature review of the PubMed database using terms of “breast” AND (“neuroendocrine carcinoma” OR “neuroendocrine tumors” OR neuroendocrine differentiation”) within the last 10 years, which were available of abstract, on December 2021. Due to the limited evidence on treatment strategies, we conducted a hand search and reviewed additional literature (Figure 1).

Point 2: Furthermore, another recent paper on the topic published in October 2021 was not cited by the authors (https://pubmed.ncbi.nlm.nih.gov/34728787/ ).

Response 2: Thank you for the important comment. We have also cited the paper which the reviewer mentioned.

Point 3: In addition, it is not clear if the images reported in the manuscript are from the research group of the authors or if they were obtained from the literature, as the authors have not even cited their articles published on this topic.

Response 3: I appreciate the comment. The figure 2 in revised manuscript was original figure by authors, and figure 3,4 (pathological pictures) were obtained by authors, which are not published in any other paper.

Point 4: In my opinion the manuscript should be improved taking into consideration that a review paper must be updated with the latest articles published on the topic and the authors should explain what is the novelty of their manuscript in light of the similar articles mentioned above.

Response 4: I sincerely appreciate the reviewer’s thoughtful comment. We agree the reviewer’s comment and have added updated information and original figures in the management section. We have added the information on management, surgery, chemotherapy, hormonal therapy, anti-HER2 therapy, somatostatin analogue, peptide receptor radionuclide therapy (PRRT) and other agent section. We focus on the possibility of clinical adaptation of novel treatment strategy or drugs developed in IBC-NST into Br-NENs. For example, anti-HER2 agent, ADCs or immune checkpoint inhibitors. Also Figure 5 and Figure 6 (by authors) were added to show overview of the treatment strategy of Br-NEN.

Point 5: In addition, the authors should specify if they have contributed data published by their group to the bibliographic review reported in the paper.

Response 5: Thank you for the important comment. We do not have contributed data in this manuscript. All figures were written by authors.

We hope that our revised manuscript will be considered for publication in the Cancers. We look forward to hearing from you in due course.

Yours sincerely.

Yukinori Ozaki

Round 2

Reviewer 1 Report

Authors address all the modification required. 

There are some errors to correct. i.e. at page 5 line 191 athors reported: Marco et al. recently reported...the reference is Gallo M et al . Please correct it

In the ICIs paragraph please include a more appropriate reference about the activity and safety of ICI in NENs DOI: 10.3390/ph14050476

Author Response

Response to Reviewer 1 Comments

Point 1: There are some errors to correct. i.e. at page 5 line 191 authors reported: Marco et al. recently reported...the reference is Gallo M et al . Please correct it

Response 1: I sincerely appreciate the reviewer’s important comment. We have corrected the error, as shown below.

“Gallo M. et al. recently reviewed the mammographic findings in case reports and case series of Br-NENs and reported that the most common mammographic appearance was a hyperdense, irregularly shaped solitary mass without calcifications (23).”

Point 2: In the ICIs paragraph please include a more appropriate reference about the activity and safety of ICI in NENs DOI: 10.3390/ph14050476

Response 2: Thank you for the important comment. We have added the review paper as a reference and sentence, as shown below in the ICIs paragraph.

“Phase 1/2 studies were conducted to evaluate the role of immune checkpoint inhibitors in NENs, which showed promising efficacy and manageable toxicity (77).”

We hope that our revised manuscript will be considered for publication in the Cancers. We look forward to hearing from you in due course.

Yours sincerely.

Yukinori Ozaki

Reviewer 3 Report

The authors have fully addressed all concerns and suggestions provided to improve the manuscript. In my opinion, the revised manuscript is now suitable for publication in present form.

Author Response

Thank you for your supportive comment. 

We appreciate all your effort to review our manuscript.

Yours sincerely.

Yukinori Ozaki